# Study of the Arrhythmogenic Profile of Dogs with Myxomatous Mitral Valve Disease in Stages B1 and B2

**DOI:** 10.3390/vetsci11100467

**Published:** 2024-10-01

**Authors:** Beatriz Almeida Santos, Jaqueline Valença Corrêa, Carolina Dragone Latini, Miriam Harumi Tsunemi, Angélica Alfonso, Luiz Henrique de Araújo Machado, Maria Lucia Gomes Lourenço

**Affiliations:** 1School of Veterinary Medicine and Animal Science, São Paulo State University (Unesp), Botucatu 18618-681, Brazil; beatriz.a.santos@unesp.br (B.A.S.); jaqueline.v.correa@unesp.br (J.V.C.); caca.dratini@gmail.com (C.D.L.); alfonso_angelica@ymail.com (A.A.); henrique.machado@unesp.br (L.H.d.A.M.); 2Institute of Biosciences, São Paulo State University (Unesp), Botucatu 18618-689, Brazil; m.tsunemi@unesp.br

**Keywords:** mitral valve disease, arrhythmogenesis, sympathovagal homeostasis

## Abstract

**Simple Summary:**

Myxomatous mitral valve disease (MMVD) is the most common cardiac disease in dogs. It affects the mitral valve apparatus and progresses slowly with age. In some breeds, especially small dogs, i.e., Cavalier King Charles Spaniels, the disease seems to progress faster than in others. According to the guidelines published by the American College of Veterinary Internal Medicine, canine MMVD can be categorized into four stages. Dogs that are affected by MMVD but have not developed clinical signs of heart failure belong to stage B. This stage is further divided into stages B1 and B2, based on the dimensions of the left-sided cardiac chambers. Specifically, stage B1 is assigned if the dimensions of the left-sided cardiac chambers are not remodeled enough to require medical treatment. When they become enlarged enough to need therapy, the disease progresses to stage B2. Although previous studies have investigated electrocardiographic abnormalities and disturbances in autonomic balance in canine MMVD, information in that regard is still limited, especially in the preclinical stages of the disease. Therefore, the present study investigated whether an increase in the dimensions of the left-sided cardiac chambers in dogs with preclinical MMVD (stage B2) would predispose them to cardiac arrhythmias and/or autonomic imbalances. A comparison of electrocardiographic findings from stage B1 and stage B2 dogs showed no statistically significant predisposition to arrhythmias in subjects with enlarged left-sided cardiac chambers. Moreover, the autonomic balance remains adequate in stage B2, as well as in stage B1.

**Abstract:**

Myxomatous mitral valve disease (MMVD) is the most prevalent cardiac disease in dogs. This study aimed to compare the arrhythmogenic profile and heart rate variability (HRV) of dogs with MMVD in stages B1 and B2. Electrocardiographic exams and the medical records of 60 dogs were analyzed, and HRV, P wave dispersion, QT interval dispersion, and QT interval instability parameters were determined. The results showed significantly increased values in stage B2 compared with stage B1 (*p* < 0.05) regarding P wave maximum and minimum duration (Pmax and Pmin) and short-term instability (STI). In contrast, no statistically significant differences were observed regarding HRV parameters, P wave dispersion, or QT interval dispersion. Our findings showed that cardiac remodeling in stage B2 could not significantly alter the sympathovagal balance and showed little interference with the predisposition of arrhythmias in dogs with MMVD.

## 1. Introduction

Myxomatous mitral valve disease (MMVD) is a chronic disease that affects between 30% and 70% of the canine population over 10 years of age, and is the most prevalent heart disease in the species [1]. In the early stages, MMVD can behave as a benign disease throughout the life of the animal [2]. However, progressive valvular degeneration leads to worsening volume overload that promotes sympathetic tone and activates the renin–angiotensin–aldosterone system over time [3]. In addition, tissue fibrosis occurs, possibly promoting the formation of substrates for arrhythmia development [4]. P wave dispersion is an electrocardiographic measurement of the duration of atrial depolarization (P wave) in multiple leads [5]. This is an important marker of atrial remodeling and a predictor of atrial fibrillation, and it involves measuring the P wave at its longest and shortest duration. In humans, P wave dispersion reflects failures in the homogeneity of sinus impulse propagation in the atria, as well as prolonged interatrial and intra-atrial conduction; characteristics that are present in patients with supraventricular arrhythmias [6].

While P wave dispersion refers to the electrophysiological properties of atrial tissue [5], QT interval dispersion is a marker of homogeneity in ventricular repolarization [7]. In humans, greater QT interval dispersion is associated with a greater predisposition to sudden death, the development of severe ventricular arrhythmias, hypertrophic cardiomyopathy, long QT syndrome, myocardial infarction, and congestive heart failure (CHF) [7]. In turn, QT interval instability is a marker for predicting ventricular repolarization instability. It predicts long QT syndrome and Torsades de Pointes [8]. To measure QT interval instability, a sequence of consecutive beats is chosen, and QT intervals are measured. Then, a center of gravity and a rotated center of gravity are calculated. The distance to the X coordinate represents the long-term instability (length of the plot), and the distance to the Y coordinate represents the short-term instability (width of the plot). Both short-term instability (STI) and long-term instability (LTI) medians are calculated. The median of the distances of all points to the center of gravity is the total instability [8].

Heart rate variability (HRV) corresponds to differences between ventricular depolarization intervals and can be measured using an electrocardiogram [9]. Increases in sympathetic tone increase the heart rate, whereas an increase in parasympathetic tone causes it to slow down. Thus, analyzing HRV in dogs allows us to assess the autonomic nervous system state, as described in human medicine [10]; this factor is reduced in heart disease and heart failure as a reflection of the increased sympathetic tone and decreased parasympathetic tone [11].

HRV assessment in dogs is possible through analysis in the time and frequency domains, as described in human medicine. Time domain analyses are measured using intervals derived from depolarizations that obtain normal-to-normal depolarization intervals (NNs), from which measurements are performed. Frequency domain analysis uses spectral analysis and Fourier transformation, transforming time domain information into the frequency domain [12]. In time domains, the SDNN (standard deviation of RR intervals) is the most commonly used parameter for assessing HRV [12] and represents the activity of the autonomic nervous system, reflecting sympathetic and parasympathetic modulation. Other parameters include rMSSD (square root of the mean square of differences between consecutive RRs) and pNN50 (the division of NN50, the number of differences in consecutive RR intervals longer than 50 milliseconds, by the number of RR intervals) reflecting parasympathetic activity [12,13].

In frequency domain components, high frequency (HF) is a marker of parasympathetic tone. In dogs, low frequency (LF) can be considered sympathetic activity and/or a set of sympathetic and vagal activities [14], in addition to baroreflex sensitivity [12], as described in humans. Therefore, the ratio between LF/HF represents the relationship between baroreflex sensitivity and parasympathetic modulation [12].

Because changes in myocardial structure and HRV reduction in dogs with MMVD predispose them to arrhythmias, the present study aims to compare the associated markers of P-wave and QT interval dispersion, QT interval instability, and HRV analyses in asymptomatic MMVD dogs with and without cardiac remodeling that requires treatment (stages B2 and B1, respectively). We will thus assess whether structural cardiac alterations, in the absence of heart failure, predispose dogs to arrhythmias, conduction disturbances, and increased sympathetic stimulation, as previously documented by other authors in dogs with CHF [11,15].

Although some studies have already presented useful data about HRV analysis [11] and QT instability [15] in dogs with MMVD, to the best of our knowledge, ours is the first to include HRV analysis, P wave and QT wave dispersion, and QT instability in evaluating dogs with MMVD.

## 2. Materials and Methods

This study was conducted according to animal welfare standards after protocol approval from the Ethics Committee on the Use of Animals—CEUA (protocol 0165/2021).

This study was conducted retrospectively with data obtained from medical records. Dogs were classified according to the criteria described in guidelines published by the American College of Veterinary Internal Medicine (ACVIM) in 2019 [16] (echocardiographic changes compatible with MMVD, systolic murmur in the mitral focus of grade ≥III/VI, a left atrium/aorta ratio of ≥1.6, and a normalized left ventricle diastolic diameter of ≥1.7) [16]. Both groups included dogs submitted to anamnesis, clinical examination, echocardiographic, and electrocardiographic exams; the latter being reanalyzed by a single evaluator to maintain standardization. This evaluator is a veterinarian, a member of the postgraduate program in Veterinary Medicine in the Cardiology Department (a veterinary with a residency in Small Animall Medicine and two years of experience in electrocardiography). Regarding drug administration, none of the enrolled dogs underwent cardiovascular therapy for MMVD treatment during the electrocardiographic analysis. In the case of stage B1 dogs, this was because no medical therapy is recommended at this stage [16]. In the case of stage B2 dogs, this was because electrocardiographic data were acquired at the first presentation, before pimobendan was prescribed [16]. Exclusion criteria were as follows: dogs with MMVD in stages other than B1 and B2; dogs with cardiac diseases other than MMVD; dogs with clinically relevant systemic diseases (e.g., infectious diseases, presence of moderate-to-severe azotemia, i.e., a serum creatinine level of >2.8, according to International Renal Interest Society classification [17]); and dogs receiving vasoactive drugs and/or sedatives at the time of data collection.

Computerized electrocardiographic exams were recorded in 3 min. The exam was performed in unsedated dogs positioned and restrained in right lateral recumbency, following the technique described by Tilley in 1992 [18]. The electrodes were attached to the humerus–radius–ulna and femur–tibia–patella joints, and damped in alcohol 70% solution. The bipolar (DI, DII, and DIII) and unipolar (aVR, aVL, and aVF) leads were recorded and processed by TEB ECG-PC VET® software. Standard electrocardiography measurements (P wave duration and amplitude, PR interval duration, QRS duration and amplitude, QT interval duration, T wave amplitude and RR intervals) were measured in a single beat and analyzed according to Santilli et al. [19].

Data collected via short-term electrocardiogram for HRV analysis were processed in the Kubios HRV Standard software (Kubios HRV 3.1, Kubios, Kuopio, Finland). After processing, analyses were performed in the time and frequency domains (SDNN, rMSSD, pNN50, LF, HF, LF/HF ratio), and the mean indices of the RR intervals (mean RR) and heart rate (mean HR) were obtained [20]. For each animal, the P wave and QT interval durations were obtained in 6 leads (D1, D2, D3, aVR, aVL, and aVF) and then the maximum and minimum durations among these values were obtained. The dispersion was calculated using the formula d = max-min (Pd = Pmax-Pmin and QTd = QTmax − QTmin) [21]. QT measurements were corrected (QTC) with TEB ECG-PC VET^®^ using Bazzet’s formula: QTC = QT/√RR. From the previous measurements, the centers of gravity and rotated centers of gravity were calculated, and in sequence, the QT mean (QTa), QT variance (QTv), total instability (TI), long-term instability (LTI), and short-term instability (STI) were determined according to the literature (2005) [8].

HRV was measured by manually counting RR intervals (in milliseconds) in lead D2 in 3 min using a TEB ECG-PC VET^®^ device. These measurements were later tabulated in Microsoft Excel and transferred to Kubios. The data were processed using this software, providing tachograms, scatter plots, and frequency graphics for analysis and SDNN, rMSSD, pNN50, LF, HF, and LF/HF indices.

The Shapiro–Wilk test evaluated the assumption of normality in the distribution of variables in the groups of dogs with stage B1 or stage B2 MMVD. The *t*-test for independent samples determined the variables in which the assumption of normality was met in the comparison between groups B1 and B2. The nonparametric Mann–Whitney test was used for variables in which the assumption of normality was not met. A *p*-value of less than 0.05 indicated a statistical difference between the groups.

Because this study aims to evaluate the arrhythmogenic profile of B-stage dogs, all of the arrhythmic events observed during the monitorization in both groups were described.

## 3. Results

The study population comprised 60 animals of both sexes (63.3% females and 36.7% males in group B1; and 43.3% females and 56.7% males in group B2). This comprised 26 mixed-breed dogs, 7 Poodles, 6 Dachshunds, 4 Lhasa Apsos, 4 Yorkshire Terriers, 3 Miniature Pinschers, 2 Maltese, 2 Shih Tzus, 2 Schnauzers, 1 Australian Cattle Dog, 1 Basset Hound, 1 Jack Russel Terrier, and 1 Labrador Retriever. Ages ranging from 1.8 to 17 were evaluated; and 30 dogs in stage B1, and 30 in stage B2 were selected. There was no statistically significant difference between the ages of the groups listed in Table 1.

When we consider standard electrocardiography measurements from a single beat [20] and the HR measured on the TEB ECG-PC VET® device, only the QRS amplitude in mV showed a statistical significance between groups (Table 2 and Table 3), showing higher values in group B2.

The base rhythm was sinus arrhythmia in most patients (25 dogs in group B1 and 24 dogs in group B2), with sinus rhythm occurring in 5 dogs from group B1 and 6 from group B2. Six dogs from group B1 and seven dogs from group B2 had sinus arrests; two dogs from group B1 had ventricular premature complexes; one dog from group B1 and five from group B2 had supraventricular premature complexes; and one dog from group B1 had a right bundle branch block and two dogs from group B2 had interatrial and/or intra-atrial blocks [19].

The means of the maximum and minimum P wave duration (Pmax and Pmin) and P wave dispersion (Pd) measurements were higher in group B2 than B1; however, a statistically significant difference (*p* < 0.05) was observed only in the parameters of minimum and maximum P wave duration (Pmin and Pmax). No significant differences were found between QT interval dispersion parameters. Regarding QT dispersion measurements, although group B2 presented higher QTmax and QTmin means, statistically, there was no significant difference between the groups in Table 4.

Except for the mean and median of the LTI values calculated with the uncorrected QT values, all the means and medians of the measurements related to the QT instability parameters were higher in group B2 than in group B1. However, a statistically significant difference (*p* < 0.05) was observed only in the STI parameters performed with uncorrected QT measurements, as depicted in Figure 1 and Figure 2, and Table 5.

Regarding HRV parameters, the RR interval means and medians were higher in group B1 than in group B2. As the intervals between beats increase, the heart rate (HR) value decreases; thus, group B1 had a lower HR than group B2. Despite some mild discrepancies in the means and medians, it was not possible to observe any statistically significant difference related to HRV parameters (*p* < 0.05) (Table 6).

## 4. Discussion

Although the electrocardiographic P wave duration values evaluated in lead D2 showed no significant difference between the groups, the Pmax and Pmin values used to calculate the P wave dispersion were higher in group B2 than B1 (*p* = 0.0140 and 0.0256, respectively), possibly because of the greater left atrial overload faced by group B2. However, in isolation, P wave duration is a limited parameter for evaluating atrial enlargement [22].

Regarding the QT interval, there was no significant difference between the dispersion indices; however, in the instability parameters, it was possible to observe an increase in STI in B2 patients compared with B1 patients (*p* = 0.0473). In the Poincaré plot for instability analysis, the STI is demonstrated by width measurements, the LTI is demonstrated by length measurements, and the TI is related to both measurements. The QT interval instability index is a mathematical model that measures variation in this interval with each heartbeat; this is related to aspects of Poincaré graphs, which dynamically depict the behavior of intervals at each beat. In this type of chart, width is a measure of short-term instability (STI), length is a measure of long-term instability (LTI), and a measure called total instability (TI) depends on both [23].

Therefore, the increase in STI shows that cardiac remodeling interferes, albeit subtly, with ventricular electrical activity, providing us with information about the onset of instability changes in all parameters, in stage C of this disease. These findings corroborate the literature when compared with a recent study (2018) that evaluated QT interval instability in MMVD patients, showing a progressive increase in the indices as the disease progressed [15]. Therefore, the QT interval instability index can quantify variation in the cardiac repolarization time, which predisposes dogs to ventricular arrhythmias [23].

Regarding the HRV parameters in the present study, it was not possible to prove a significant difference between the groups, although the averages of the SDNN, rMSSD, and pNN50 parameters were slightly higher in group B1, and there was a higher HF average and lower LF/HF ratio in the same group. The HRV analysis values for both groups were similar to those observed by Martinello et al. (2022) in healthy animals of similar ages [20]. This allows us to infer that, alone, cardiac remodeling in MMVD cannot change the sympathetic–parasympathetic balance in dogs without CHF. The same can be observed in analyses of dogs with CHF secondary to MMVD, such as in the study by Oliveira et al. (2012), which observed a reduction in SDANN and pNN50 parameters in a group of animals in stage C [11].

These findings suggest that the changes in sympathetic tone detected in patients with CHF are not present in the initial stages of cardiac remodeling. In another recent study, a reduction in HRV parameters was observed in the time domains, in relation to stages C and B but without differences between two animals labeled B1 and B2. However, in the same study, the frequency domain analysis showed a difference between animals B1 and B2, showing a reduction in the HF vagal component and an increase in the LF/HF ratio in B2 in relation to B1 [24].

Some of the limitations of this study included the difficulty of standardizing information previously obtained about the animals, as the animals were not clinically evaluated or examined by the same veterinarians. Additionally, considering that some of the animals first presented at the Cardiology Service, it was difficult to obtain a complete medical history of comorbidities from the veterinarian’s reports. Because the present study used information previously obtained from patients without sedation, some exams presented many artifacts, making it difficult to measure a series of consecutive parameters. Additionally, most of the animals were not familiar with the hospital environment, a factor that can generate stress and, therefore, changes in physiological parameters.

Another limitation regarding the retrospective nature of this study is that only a few patients had fully cardiologic standardized evaluations that were sufficient to fulfill the inclusion criteria; for that reason, only a few were included. This is also why creating a control group and an analysis of larger ECG tracings (i.e., 24-h Holter monitorization) were not possible, as healthy patients usually do not undergo a complete cardiologic standardized evaluation.

Another limitation is that the electrocardiographic tracings were reviewed by one examiner, leading to a lack of inter-observer analysis. The lack of statistics for the echocardiographic exams is also a limitation.

Due to the fact that this study focused on more specific analysis, some standard ECG parameters were not statistically evaluated, such as the mean electrical axes of P wave, the ST segment amplitude, the T wave duration and mean electrical axes of the QRS complex.

Lastly, regarding stage B2 patients, it is possible to observe a wide range of phenotypes, ranging from dogs with signs of cardiac remodeling to dogs about to present congestive heart failure (the latter are patients evolving to stage C). The different sizes of cardiac chambers, as well as the different hemodynamic statuses, make B2 patients a group where electrophysiological changes may manifest in different ways as the disease progresses. Therefore, to precisely determine when alterations begin, further studies are needed to subclassify the B2 group and correlate electrophysiological changes to the severity of the disease.

## 5. Conclusions

The reduction in HRV indices described in the literature in patients with stage C MMVD was not observed to the same extent in patients with stage B MMVD. The reasons for this also apply to P wave and QT interval dispersion parameters; however, in contrast, when analyzing QT interval instability, it is possible to observe an increase in STI, demonstrating the beginning of changes in the homogeneity of ventricular electrical activity often reported in later stages.

## Figures and Tables

**Figure 1 vetsci-11-00467-f001:**
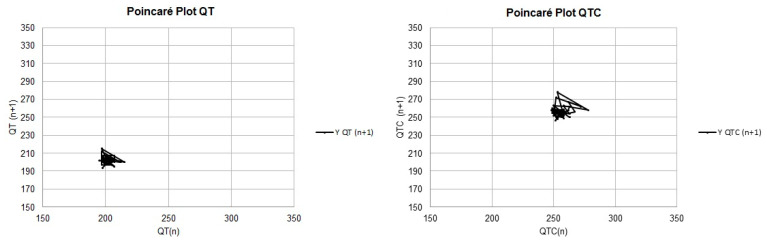
Modified Poincaré plots, with a line connecting the successive QT and QTC interval measurements, demonstrating QT instability in patient B1.

**Figure 2 vetsci-11-00467-f002:**
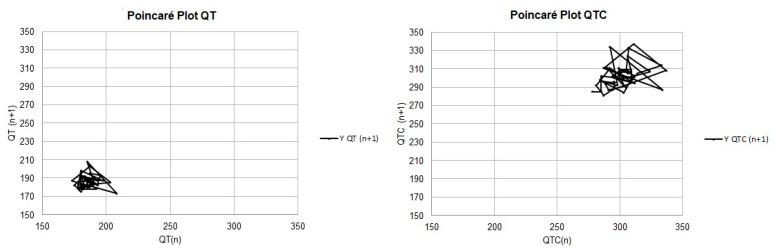
Modified Poincaré plots, with a line connecting the successive QT and QTC interval measurements, demonstrating QT instability in patient B2.

**Table 1 vetsci-11-00467-t001:** Characterization of groups in dogs with myxomatous mitral valve disease in stages B1 and B2.

Variables	B1	B2	*p*-Values
Age	10.5	11.6	0.193
Males	11 (36.7%)	17 (56.7%)	0.1957
Females	19 (63.3%)	13 (43.4%)	0.1957

**Table 2 vetsci-11-00467-t002:** Electrocardiography standard measurements from a single beat in dogs with myxomatous mitral valve disease in stages B1 and B2.

Parameters	Mean	SD	Reference Intervals [20]	Minimum	Median	Maximum	*p*-Values
HR (B1)	124.000	25.112	60–170 bpm	78.000	123.500	169.000	0.182
HR (B2)	123.967	25.173	60–170 bpm	79.000	124.000	174.000	0.608
P (ms) (B1)	52.367	4.679	<40 ms	43.000	52.500	62.000	0.448
P (ms) (B2)	53.033	6.139	<40 ms	43.000	52.000	70.000	0.313
P (mv) (B1)	0.231	0.088	<0.4 mV	0.058	0.224	0.461	0.874
P (mv) (B2)	0.263	0.114	<0.4 mV	0.070	0.248	0.547	0.032
PR (ms) (B1)	88.800	13.627	60–130 ms	62.000	89.500	112.000	0.567
PR (ms) (B2)	90.000	15.148	60–130 ms	62.000	88.500	123.000	0.035
QRS (ms) (B1)	56.433	7.500	<70 ms	45.000	57.000	73.000	0.261
QRS (ms) (B2)	58.567	6.135	<70 ms	45.000	58.000	73.000	0.972
QRS (mV) (B1)	1.060	0.402	<3 mV	0.125	1.068	1.820	0.995
QRS (mV) (B2)	1.604	0.664	<3 mV	0.410	1.586	3.242	0.595
QT (ms) (B1)	190.700	16.449	150–240 ms	167.000	187.000	237.000	0.111
QT (ms) (B2)	191.767	15.281	150–240 ms	170.000	190.000	230.000	0.284
T (mV) (B1)	−0.132	0.290	<0.05–1 mV (+ or −)	−0.715	−0.146	−0.781	0.016
T (mV) (B2)	−0.106	0.339	<0.05–1 mV (+ or −)	−0.733	−0.182	0.672	0.324

**Table 3 vetsci-11-00467-t003:** ECG statistics in dogs with myxomatous mitral valve disease in stages B1 and B2.

Parameters	*p*-Value
P (ms)	0.6380
P (mV)	0.2970
PR (ms)	0.9764
QRS (ms)	0.2327
QRS (mV)	0.0004 *
QT (ms)	0.7956
T (mV)	0.9823

* = *p*-value < 0.05 (*t*-test).

**Table 4 vetsci-11-00467-t004:** P wave dispersion and QT interval indices in dogs with myxomatous mitral valve disease in stages B1 and B2.

	B1	B2	
Parameters ^1^	Mean/SD	Mean/SD	*p*-Values
P max	55.46 ± 4.75	59.03 ± 6.07	0.0140 *
P min	41.33 ± 5.66	44.53 ± 5.14	0.0256 *
Pd	14.13 ± 6.76	14.50 ± 5.75	0.5280
QT max	199.63 ± 18.48	203.60 ± 17.62	0.3984
QT min	183.43 ± 18.83	187.43 ± 19.10	0.4175
QTd	16.20 ± 10.14	16.16 ± 5.71	0.3766

^1^ P max (maximum P wave duration), P min (minimum P wave duration), Pd (P wave dispersion), QT max (maximum QT interval duration), QT min, (minimum QT interval duration), and QTd (QT interval dispersion). * *p*-value < 0.05 (*t*-test).

**Table 5 vetsci-11-00467-t005:** Analysis of QT interval instability in dogs with myxomatous mitral valve disease in stages B1 and B2.

	B1	B2	
Parameters ^1^	Mean/SD	Mean/SD	*p*-Values
QTa	189.33 ± 16.27	190.01 ± 15.94	0.869
QTv	15.71 ± 14.46	32.99 ± 61.71	0.358
TI	6.59 ± 1.30	7.34 ± 2.59	0.321
LTI	5.57 ± 0.94	5.56 ± 1.22	0.615
STI	2.16 ± 1.03	3.10 ± 2.50	0.047 *
QTca	271.35 ± 18.28	275.58 ± 19.76	0.393
QTcv	169.56 ± 130.02	202.23 ± 147.40	0.423
TIc	15.40 ± 5.29	16.22 ± 6.01	0.580
LTIc	11.47 ± 3.57	12.18 ± 4.26	0.686
STIc	6.16 ± 3.51	6.82 ± 3.81	0.544

^1^ QTa (mean of QT intervals), QTv (variance of QT intervals), TI (total instability), LTI (long-term instability), STI (short-term instability), QTca (mean of corrected QTs), QTcv (variance of corrected QTs), TIc (total instability of corrected QTs), LTIc (long-term instability of corrected QTs), and STIc (short-term instability of corrected QTs). * = *p*-value < 0.05 (*t*-test).

**Table 6 vetsci-11-00467-t006:** Heart rate variability assessment in the time and frequency domains in dogs with myxomatous mitral valve disease in stages B1 and B2.

	B1	B2	
Parameters ^1^	Mean/SD	Mean/SD	*p*-Values
Mean RR	492.83 ± 88.98	487.83 ± 95.06	0.6573
Mean HR	125.90 ± 23.27	127.10 ± 22.47	0.8397
SDNN	70.76 ± 42.17	70.39 ± 44.30	0.9882
rMSSD	98.60 ± 75.93	94.00 ± 75.96	0.8130
pNN50	43.25 ± 26.56	38.84 ± 23.41	0.4976
LF	34.81 ± 17.49	36.95 ± 21.35	0.7412
HF	64.68 ± 17.26	62.59 ± 21.16	0.6761
LF/HF	0.70 ± 0.72	0.85 ± 0.93	0.7349

^1^ Mean RR (mean of RR intervals), mean HR (mean heart rate), SDNN (standard deviation of means of RR intervals), rMSSD (mean square root of differences between consecutive RR), pNN50 (number of consecutive RR intervals greater than 50 ms), LF (low frequency), HF (high frequency), and LF/HF (low frequency–high frequency ratio). * = *p*-value < 0.05 (*t*-test).

## Data Availability

The data from this study are stored in the University Repository Unesp (link https://repositorio.unesp.br/server/api/core/bitstreams/ef13ba61-f3e7-4044-a553-78af16d0414b/content) accessed on 27 September 2024.

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
