# Peer review of "Study of the Arrhythmogenic Profile of Dogs with Myxomatous Mitral Valve Disease in Stages B1 and B2"

_vetsci, 2024, doi:10.3390/vetsci11100467_

Round 1

Reviewer 1 Report

Comments and Suggestions for Authors

I thank the Editor for considering me for the review of this interesting manuscript. The authors' study idea is intriguing, as they aim to investigate a topic in arrhythmology that, to date, has been scarcely studied in dogs. However, there are many changes that should be made to improve the current version of the manuscript, including improving the clarity of some sentences, providing additional data, introducing further references, and performing more statistical/comparative analysis. Below some corrections, suggestions and indications that I hope could be useful for readers.

Simple summary

-“Myxomatous mitral degenerative valve disease is a common condition in small breed dogs, being one of the predominant in the clinical scenario of cardiomyopathies in dogs.”

I suggest to simple state that it is the most common cardiac disease in dogs. Moreover, I would not write “Myxomatous mitral degenerative valve disease” if you then wanto to cite tricuspid degeneration. I try to explain myself better. In other terms, if in the following statement you pwant to cite the possible involvement of tricuspid valve, in the initial sentence you should use the term “Myxomatous degenerative valve disease” without the word “mitral”. In contrast, if you prefer to use the complete terminolgy in the initial sentence (i,e., the one that includes the term “mitral” = “Myxomatous mitral degenerative valve disease”), then you should not cite tricuspid valve in the following statement. Moreover, I suggest to introduce at the beginning an abbreviation (e.g., MMVD) and then use that.

-“It is staged in four stages; when it comes to stage B, it is an intermediate phase of the disease, where cardiac remodeling occurs (B2).”

This statement is imprecise. I suggest to rewrite it in this way or something similar: “According to the guidelines published by the American College of Veterinary Internal Medicine, canine MMVD is categorized into four stages. Dogs that are affected by MMVD but have never developed clinical signs of heart failure belong to stage B. This stage is further divided into stage B1 and stage B2 based on the dimensions of the left-sided cardiac chambers. Specifically, stage B1 is assigned if the dimensions of the left-sided cardiac chambers are within normal limits; when they become enlarged, the disease progresses to stage B2."

“The present study investigated whether the increase in cardiac silhouette (stage B2) would provide a greater predisposition to cardiac arrhythmias. The results showed that there was no predisposition to arrhythmias during this stage of the disease, and the autonomic balance remains adequate.”

There are inaccuracies in this section. First, if the aim of the study is to explore arrhythmias/ECG abnormalities in stage B dogs, you should first establish the link between MMVD and arrhythmias (so that readers can understand why you and set up this specific type of study). Second, the term '’silhouette’' is generally limited to radiology; in a study like the present one, it is preferable to use the term '’increased dimensions of the left-sided cardiac chambers.’' Here, a revised version of these sentences: ‘'Although some previous studies investigated the presence of electrocardiographic abnormalities and disturbances of autonomic balance in canine MMVD, information at regard are still limited in dogs, especially in the preclinical stages of the disease. Therefore, the present study aimed at investigating whether an increase in the dimensions of the left-sided cardiac chambers in dogs with preclinical MMVD (stage B2) would predispose dogs to cardiac arrhythmias and/or autonomic imbalnces. The comparison of electrocardiographic findings from stage B1 and stage B2 dogs showed no statistically significant predisposition to arrhythmias in subjects with enlargmente of left-sided cardiac chambers. Moreover, the autonomic balance remains adequate in stage B2 as well as in stage B1.'’

Abstract

-line 22: “…dogs with MMVD in stages B1 and B2; Electrocardiographic tracings…”. There is a typing error. The correct version is: “…dogs with MMVD in stages B1 and B2. Electrocardiographic tracings…”.

-“The results showed significantly increased values in group B2 (p <0,05) in the parameters of P wave maximum and minimum duration (Pmax and Pmin) and short-term instability (STI). No significant difference was observed in HRV parameters, P wave dispersion or QT interval dispersion.”

 I suggest to modify this sentence in this way: “The results showed significantly increased values in stage B2 compared to stage B1 (p <0.05) regarding P wave maximum and minimum duration (Pmax and Pmin) and short-term instability (STI). In contrast, no statistically significant differences were observed regarding HRV parameters, P wave dispersion and QT interval dispersion.”

-“The heterogeneity present in the B2 stage includes dogs with different sizes of cardiac chambers and hemodynamic status. Therefore, further studies are required to subclassify those patients and, determine the beginning of the sympathovagal imbalance and predisposition of arrhythmias, both widely described in stage C of the disease.”

This part should be deleted, as it is sounds more like a part of the discussion. It is not essential in the abstract section.

Introduction

-“Progressive degeneration, however, leads to insufficiency of the valve apparatus and consequent volume overload that promotes,over time, activation of sympathetic tone and the renin angiotensin aldosterone system [3], in addition to lesions such as tissue fibrosis, which promotes the formation of substrates for the development of arrhythmias [4].”

I suggest to break this sentecne in two part as it is very long: “However, progressive valvular degeneration leads to worsening volume overload that promotes, over time, activation of sympathetic tone and the renin angiotensin aldosterone system [3]. In addition, tissue fibrosis occurs, which may promote the formation of substrates for the development of arrhythmias [4].”

-Concerning the rest of the Introduction, some problems are present. Here a list:

1) in the ligh of the previous sentence, I would have expected to read something about the investigation of arrhtyhmias, espcially those that are more common in MMVD (e.g., supraventricualr arrhythmias, ventricualr arrhythmias, and/or disturbances of atrioventricular conduction). In contrast, in the following sentences you described/investigated some very specific electrocardiographic parameters (not usual in veterinary medicine) (e.g., those related to P and QT instability). Why? Please explain why you wanted to investigate these very specific parameters and not to set up a more traditional anlysis (e.g., as the one done in reference n. 4) as this is not clear on the base of the current version of your introduction.

2) Some sentences referes to veterinary studies, other are from human medicine. In each sentence (here and also later in M&M and Discussion), you should specify which species you are referring to, so that readers can easily understand if the electrocardiographic parameters/findings you are describing have been already studied in dog or not and, in the second case, if they are taken from human literature.

3) Since there are already some studies published on dogs concerning the ECG parameters you investigated, you should clearly explain the additional advantage your study offers. In other words, you should specify which gap in the current scientific literature your study aims to fill.

4) You use some abbreviation directly (e.g., “STI”, “LTI” and others). Please, correct each time you made this error.

5) You should explain, providing a scientifically acceptable reason, why you specifically investigated these ECG parameters in stage B dogs and not also in other stages (also considering that other stages, such as C and D, could have been more interesting by an arrhythmologic point of view).

6) line 94: “…as seen in dogs with congestive heart failure (CHF).” This should be rewritten as: “… as previously documented by other authors in dogs with congestive heart failure (CHF).” Then, you should provide pertinent references.

Given the above, I think that the introduction of this study should be significantly modified by Authors.

M&M

-“…in which 60 dogs with MMVD of both sexes, different breeds and ages ranging from 1.8 to 17 years were evaluated, and 30 dogs in stage B1 and 30 dogs in stage B2 were selected.”

This is wrong: all the numbers you are reporting are, as a matter of facts, an anticipation of results. In the M&M section you should simply state your methodology. Then, in the Result section, you will provide information of the number of dogs from each stage, their age, body weight, etc…

-“Both groups were classified according to the criteria of the American College of Veterinary Internal Medicine (ACVIM) guidelines in 2018 [15] (echocardiographic changes compatible with MMVD, systolic murmur in mitral focus ≥ III/VI, left atrium/aorta ratio ≥1.6 and left ventricle diastolic diameter normalized ≥1.7).”

Please, correct rewriting in this way: “Dogs were classified according to the criteria described in guidelines published by the American College of Veterinary Internal Medicine (ACVIM) guidelines in 2019 [15] (echocardiographic changes compatible with MMVD, systolic murmur in mitral focus ≥ III/VI, left atrium/aorta ratio ≥1.6 and left ventricle diastolic diameter normalized ≥1.7). Then, at the end of this sentence, add the proper reference (Keene et al., 2019).

-“Regarding drug administration, none of the groups used cardiovascular therapy for the treatment of MMVD (animals from group B2 were first diagnosed).”

Please rewrite as: “Regarding drug administration, none of the enrolled dogs used cardiovascular therapy for the treatment of MMVD at the time of electrocardiographic analysis. In the case of stage B1 dogs, this was due to the fact that no medical therapy is recommended at this stage (Keene et al., 2019). In the case of stage B2 dogs, this was due to the fact that electrocardiographic data were acquired at the first presentation of the dogs, before the prescription of pimobendan (Keene et al., 2019). *(I hope to have correctly understood your methodology and what really happened).

-“Exclusion criteria were terminally ill patients, patients with MMVD in stages C or D, patients with dilated cardiomyopathy, arrhythmogenic right ventricular cardiomyopathy, congenital heart disease, sick sinus node syndrome, myocarditis, infectious diseases, use of vasoactive drugs and/or sedatives at the time of data collection, and presence of azotemia at levels above moderate (serum creatinine >2.8, according to IRIS classification).”

Please rewrite as: “Exclusion criteria dogs with MMVD in stages different from B1 and B2, dogs with cardiac diseases different from MMVD, dogs with clinically relevant systemic diseases (e.g., infectious diseases presence of moderate-to-severe azotemia [i.e., serum creatinine >2.8, according to IRIS classification]), and dogs receiving vasoactive drugs and/or sedatives at the time of data collection.” Additionally, the abbreviation “IRIS” is used directly, which represents a mistake. Moreover, nart to the cut-off proposed by IRIS you should use the proper reference.

-Line 116: please, introduce at least one reference. I think that references should be added also at the end of sentences at lines 122, 126 and 133. In this way, readers will know which are the references/studies on which you have based your methodologies/measurements.

-Line 117: please, specify the degree of expertise of the examiner (was an ACVIM or ECVIM cardiology Diplomate? Not a board-certified cardiologist, but he/she holds a PhD or a MS in cardiology? Neither a board-certified cardiologist nor a vet with a PhD/MS in cardiology, but he/she is anyway a vet with many years of specilistic experience in small animal cardiology? Or was he/she a general vet?).

-Lines 142-143: “This study being conducted according to animal welfare standards, after approval of the protocol by the Ethics Committee on the Use of Animals - CEUA (protocol 0165/2021).”

I think that this statment should appear at the beginning of M&M, not at its end (as usually done in other similar studies from canine cardiology).

-Additional comments:

1) Lines 123-126: “P-wave dispersion (Pd) and QT interval dispersion (QTd) measurements were obtained from each of the six recorded electrocardiographic leads (D1, D2, D3, aVR, aVL, and aVF). For each animal, the maximum and minimum durations (Pmax, Pmin, QTmax and QTmin) were obtained, and the dispersion was calculated using the formula d = max-min.”

I’m not sure to have completely/correctly understood your methodology. I read that you have mesured P and QT dispersion in each one of the six leads. Then my question is: which measurements did you considered in you analysis? Exclusively the ones from DII (as this lead is the one more used in canine cardiology)? Or did you perform a mean of values obtained in each lead? Or, alternatively, did you used simply the maximal value (regardless the lead from which it was obtained) instead of performing a mean?

2) Even if Authors have focused their analysis on the study of very specific ECG parameters, I think that it would be very important to provide also “basic”/”conventional” ECG data from the present study population (both from stage B1 dogs and stage B2 ones). Additional data should include all conventional ECG parameters, namely:

-duration, amplitude and mean electrical axis of P wave;

-duration of PQ interval;

-duration and mean electrical axis of QRS complex, and amplitude of R wave;

-presence/absence of ST segment deviation and its amplitude;

-duration of QT and QTc interval;

-duration, polarity and amplitude of T waves.

Report mean/median of each parameter for B1 and B2 dogs along with pertinent reference intervals Of note, there are many books that provide reference ranges for many parameters. For those typically not reported by books, you can use papers available int he current canine literature (e.g., P wave mean electrical axis = doi. 10.1016/j.jvc.2022.05.001 // ST segment = doi: 10.1111/jsap.13532. // T wave = doi: 10.1016/j.jvc.2022.06.003.).

Than, perform a statistical comparison of results from B1 and B2 dogs. All this will be certainly useful for readers to start to see/understand if some ECG difference exists between B1 and B2 dogs, before focus on more advanced ECG parameters.

3) There is also another lacking information. Did you noted also if, over the ECG recording, some of the enrolled dogs showed some disturbance of cardiac rhythm (e.g., a VPC, an APC or a second-degree AVB)? Or were all dogs compltely free from any arrhythmias duuring data recording? Please specify this in the manuscript: in the M&M you should specify if you noted the possibile presence of these abnormalities; in the Rsult section, you should report the numeber of dogs showing thm and which arrhythmias you identified.

Results

-“Regarding HRV parameters, the means and medians of RR were higher in group B1 than in group B2, whereas consequently, the HR values were inversely proportional.”

This sentence should be rephrase as I think that it will be very difficult to be understood by readers.

-Lines 171-177: I my opinion, it is not essential to discuss data that did not reach the statistical significance. Therefore, you can delete all statement from this paragraph apart thie last one. You can rewrite it in this way: “Despite some mild discrepancies in the values of means and medians, it was not possible to observe any statistically significant difference related to HVR parameters (p <0.05)”.

Discussion

-“ The electrocardiographic P wave represents atrial muscle activation; when prolonged in duration, it is indicative of left atrial enlargement”

There is also another differential diagnosis. Please addi it. Moreover, at least a reference should be added at the end of this sentence. I suggest a book and also this manuscript: doi: 10.1111/j.1748-5827.2012.01200.x.

-“ These findings corroborate what was observed in the literature when compared to a recent study (2018), which evaluated the instability of the QT interval in patients with MMVD and observed a progressive increase in the indices as the disease progressed 17.”

There is a typing error at the end of the sentence regardin reference n. 17.

-“Another study in 2021, which evaluated QT measurements corrected by two different methods in

patients with MMVD, observed significant differences between patients in groups B and C; however, when comparing B1 and B2, there was no significant difference. Such findings

show a progressive tendency to increase instability as the disease progresses.”

You should provide a reference, otherwise readers cannot understant which is the study published in 2021 you are referring to.

-“ The values of the HRV analysis of both groups were similar to those observed by Martinello et al. (2022) in healthy animals of similar ages, which allows us to infer that, alone, the presence of cardiac remodeling in MMVD was not able to change the sympathetic-parasympathetic balance in dogs without CHF, as observed in studies that analyzed dogs with CHF secondary to MMVD, such as the study by Oliveira et al. (2012), which observed a reduction in SDANN and pNN50 parameters in the group of animals in stage C.”

This is a very long sentence. I suggest you to rephrase it and break it into 3 different sententences to improve its clarity.

-“ In another recent study, a reduction in HRV parameters was observed in the time domains in relation to stages C and B, but without differences between animals B1 and B2. In the same study, however, the analysis in the frequency domains showed a significant difference between B1 and B2 ani239 mals, obtaining a reduction in the HF vagal component and an increase in the LF/HF ratio in B2 in relation to B120.”

There is a typing error at the end of the sentence regardin reference n. 20.

-“Although in the present study there was no statistically significant difference, it is possible to observe similar behavior in the mean values, demonstrating evidence of the transition from autonomic status to stage C”.

As yor analysis did not reach the statistical significance, you cannot state this. Please, delete this sentence.

-Additional comment for the entire Discussion: as said before, in each sentence and for each reference, it is important to specify which species you are referring to (so that readers can know wheter the findings you are describing is from canine or human literature).

Limitations

-I think that, in addition to the limitations already reported by the Authors, other relevant limitations should be listed, including: 1) the very limited number of dogs in each group; 2) the lack of a control group composed of 30 healthy dogs (to document whether B1 and/or B2 dogs have differences compared to healthy ones regarding the parameters examined by the authors); 3) the lack of intra- and inter-observer variability analysis concerning the measurements of examined advanced ECG parameters; and 4) above all, the lack of 24-hour Holter monitoring (to obtain additional and more appropriate data on HRV parameters). Indeed, it is well known that, for many technical and physiological reasons, HRV based on Holter recordings (especially if the Holter monitoring lasts for ≥ 20 hours, is free from artifacts, and includes recordings from the night) provide a more precise/realistic picture of the autonomic balance of the patient compared to the one that cane obtained by a simple ECG performed over a few minutes in the hospital setting.

-An additonal comment related to limitation is associated with the following sentence: “Regarding stage B2 patients, it is possible to observe a wide range of phenotypes, ranging from dogs with signs of cardiac remodeling to dogs about to present congestive heart failure (the latter, patients evolving to stage C). The different sizes of cardiac chambers as well as the different hemodynamic status turn B2 patients into a particular group where electrophysiological changes may manifest in different ways as the disease progresses. Therefore, to precisely determine the beginning of the alterations, further studies are needed to subclassify the B2 group and correlate electrophysiological changes to the severity of the disease”.

You included this section in the Conclusions. I believe it would be more appropriate to delete it from the Conclusions and instead discuss it in the Limitations section. Additionally, you should include among your limitations the fact that you did not provide data on the echocardiographic parameters of your dogs. These data would have been useful to address the aforementioned point. Indeed, echocardiographic data would have provided information on the severity and heterogeneity of your population of B2 dogs. This could have been achieved simply by providing data on the enlargement of left-sided cardiac chambers among B2 dogs (e.g., LA/Ao ratio, LAD, LA volume), the severity of their left ventricular filling pressures (e.g., peak E wave, E/A ratio, E/IVRT, E/E' ratio), and the possible presence and degree of post-capillary pulmonary hypertension (e.g., by measuring the peak velocity of tricuspid regurgitation).

Conclusions

-Please delete from this section the part discussed above.

-Moreover, your conclusions should be overall changed in my opinion. Primarily, your conclusions should be significantly reduced, as they are very long (also because in this section you have repeated many concepts already discussed in previous sections of the manuscript). Usually, the Conclusions section is expected to be concise (e.g., 4-5 lines) and provide a very brief summary of the main findings of the study, nothing more.

References

-I think the list should be extended, at least by adding the study cited before.

Comments on the Quality of English Language

In some sentences, editing of English language is required.

Author Response

Responses to reviewer 1:

1 - Comment: “I suggest to simple state that it is the most common cardiac disease in dogs. Moreover, I would not write “Myxomatous mitral degenerative valve disease” if you then wanto to cite tricuspid degeneration. I try to explain myself better. In other terms, if in the following statement you pwant to cite the possible involvement of tricuspid valve, in the initial sentence you should use the term “Myxomatous degenerative valve disease” without the word “mitral”. In contrast, if you prefer to use the complete terminolgy in the initial sentence (i,e., the one that includes the term “mitral” = “Myxomatous mitral degenerative valve disease”), then you should not cite tricuspid valve in the following statement. Moreover, I suggest to introduce at the beginning an abbreviation (e.g.,MMVD) and then use that”.

-  Response: Thank you for your consideration. We agree with the suggestion, revising the sentence as follows: “Myxomatous mitral valve disease (MMVD) is the most common cardiac disease in dogs.”. Also, we have used the abbreviation “MMVD” as suggested above.

____

2 - Comment: This statement is imprecise. I suggest to rewrite it in this way or something similar:“According to the guidelines published by the American College of Veterinary Internal Medicine, canine MMVD is categorized into four stages. Dogs that are affected by MMVD but have never developed clinical signs of heart failure belong to stage B. This stage is further divided into stage B1 and stage B2 based on the dimensions of the leftsided cardiac chambers. Specifically, stage B1 is assigned if the dimensions of the leftsided cardiac chambers are within normal limits; when they become enlarged, the disease progresses to stage B2."

- Response: Thank you for your considerations. We have revised the sentence accordingly, also incorporating the suggestions from the other reviewer abou t the same paragraph, that states  “Throughout the manuscript, the authors state that there is no cardiac remodeling in B1 but only in B2. I do not think this is correct. The B2 category includes heart sizes with significant cardiac enlargement to warrant pimobendan administration, not a delineation between having cardiac remodeling vs. no remodeling. In fact, there are many dogs in the B1 category with larger than normal LA but just not big enough to meet the criteria for starting pimobendan

    The final sentence was the rewritten as: “According to the guidelines published by the American College of Veterinary Internal Medicine, canine MMVD is categorized into four stages. Dogs that are affected by MMVD but have never developed clinical signs of heart failure belong to stage B. This stage is further divided into stage B1 and stage B2 based on the dimensions of the left-sided cardiac chambers. Specifically, stage B1 is assigned if the dimensions of the left-sided cardiac chambers are not remodeled enough to require medical treatment. When they become enlarged enough to need therapy, the disease progresses to stage B2.

____

3 - Comment: There are inaccuracies in this section. First, if the aim of the study is to explore arrhythmias/ECG abnormalities in stage B dogs, you should first establish the link between MMVD and arrhythmias (so that readers can understand why you and set up this specific type of study). Second, the term '’silhouette’' is generally limited to radiology; in a study like the present one, it is preferable to use the term '’increased dimensions of the left-sided cardiac chambers.’' Here, a revised version of these sentences: ‘'Although some previous studies investigated the presence of electrocardiographic abnormalities and disturbances of autonomic balance in canine

MMVD, information at regard are still limited in dogs, especially in the preclinical stages of the disease. Therefore, the present study aimed at investigating whether na increase in the dimensions of the left-sided cardiac chambers in dogs with preclinical MMVD (stage B2) would predispose dogs to cardiac arrhythmias and/or autonomic imbalnces. The comparison of electrocardiographic findings from stage B1 and stage B2 dogs showed no statistically significant predisposition to arrhythmias in subjects with enlargmente of left-sided cardiac chambers. Moreover, the autonomic balance remains adequate in stage B2 as well as in stage B1.'’

- Response: Thank you so much for pointing this out. We agree with this statement and have incorporated your suggestion by rewriting the sentence as suggested above: “Although some previous studies investigated the presente of electrocardiographic abnormalities ane disturbances of autonomic balance in canine MMVD, information at regard are still limited in dogs, especially in the preclinical stages of the disease. Therefore, the present study aimed at investigating whether an increase in the dimensions of the left-sided cardiac chambers in dogs with preclinical MMVD (stage B2) would predispose dogs to cardiac arrhythmias and/or autonomic imbalances. The comparison of electrocardiographic findings from stage B1 and stage B2 dogs showed no statistically significant predisposition to arrhythmias in subjects with enlargement of left-sided cardiac chambers. Moreover, the autonomic balance remais adequate in stage B2 as well as in stage B1.”  

___

4 - Comment: I suggest to modify this sentence in this way: “The results showed significantly increased values in stage B2 compared to stage B1 (p <0.05) regarding P wave maximum and minimum duration (Pmax and Pmin) and short-term instability (STI). In contrast, no statistically significant differences were observed regarding HRV parameters, P wave dispersion and QT interval dispersion.”

- Response: Thank you for your suggestion. We have rewritten the sentence as suggested. “The results showed significantly increased values in stage B2 compared to stage B1 (p <0,05) regarding P wave maximum and minimum duration (Pmax and Pmin) and short-term instability (STI). In contrast, no statistically significant differences were observed regardind HRV parameters, P wave dispersion and QT interval dispersion”.

___

5 - Comment: This part should be deleted, as it is sounds more like a part of the discussion. It is not essential in the abstract section.

- Response: Thank you for pointing this out. We agree with this statement and have deleted the sentence.

___

6 – Comment: I suggest to break this sentecne in two part as it is very long: “However, progressive valvular degeneration leads to worsening volume overload that promotes, over time, activation of sympathetic tone and the renin angiotensin aldosterone system [3]. In addition, tissue fibrosis occurs, which may promote the formation of substrates for the development of arrhythmias [4].”

- Response: We agree with the statemente above, and divided the sentence in two parts as suggested: “However, progressive valvular degeneration leads to worsening volume overload thar promotes, over time, activation of sypathetic tone and the renin angiotensin aldosterona system [3]. In addition, tissue fibrosis occurs, which may promote the formation of substrates for the develompent of arrhythmias [4].”

____

7 – Comment - Concerning the rest of the Introduction, some problems are present. Here a list:

1) in the ligh of the previous sentence, I would have expected to read something about the investigation of arrhtyhmias, espcially those that are more common in MMVD (e.g., supraventricualr arrhythmias, ventricualr arrhythmias, and/or disturbances of atrioventricular conduction). In contrast, in the following sentences you described/investigated some very specific electrocardiographic parameters (not usual in veterinary medicine) (e.g., those related to P and QT instability). Why? Please explain why you wanted to investigate these very specific parameters and not to set up a more

traditional anlysis (e.g., as the one done in reference n. 4) as this is not clear on the base of the current version of your introduction.

- Response: We appreciate your appointments. Information regards arrhythmic events ane standard ECG measurements was added (as requested by multiple reviewers) in the study.

2) Some sentences referes to veterinary studies, other are from human medicine. In each sentence (here and also later in M&M and Discussion), you should specify which species you are referring to, so that readers can easily understand if the electrocardiographic parameters/findings you are describing have been already studied in dog or not and, in the second case, if they are taken from human literature.

- Response: Information about human medicine studies was added as requested, for references 6, 10, 12, 13 and 14.

3) Since there are already some studies published on dogs concerning the ECG parameters you investigated, you should clearly explain the additional advantage your

study offers. In other words, you should specify which gap in the current scientific literature your study aims to fill.

Response: Thank you for your reminder. This information was added in introduction’s section, as follows: “Although some studies have already presented precious data about HRV analisys [11] and QT instability [15] in dogs with MMVD, to this author’s knowledge, this is the first study to include HRV analysis, P wave and QT wave dispersion and QT instability in the evaluation of dogs with MMVD.

___

4) You use some abbreviation directly (e.g., “STI”, “LTI” and others). Please, correct

each time you made this error.  QTa (mean of QT intervals), QTv (variance of QT intervals), TI (total instability), LTI (long-term instability), STI (short-term instability), QTca (mean of corrected QTs), QTcv (variance of corrected QTs), TIc (total instability of corrected QTs), LTIc (long-term instability of corrected QTs), STIc (short-term instability of corrected QTs) we include in the text.

___

5) You should explain, providing a scientifically acceptable reason, why you specifically investigated these ECG parameters in stage B dogs and not also in other stages (also considering that other stages, such as C and D, could have been more interesting by an arrhythmologic point of view).

- Response: Other authors, such as DOI: 10.1016/j.jvc.2018.06.002 and DOI: 10.1136/vr.100202  already presented studies about dogs from cardiac insufficiency stages. This study aimed to investigate more about the changings in arrhythmogenic profile between dogs that are no longer in CHF. In other words, aimed  to investigate if the changings in cardiac remodeling that occurs before CHF happens would be enough to cause statistically significative changings in HRV analysis, P wave dispersion, QT wave dispersion and QTinstability.

____

6) line 94: “…as seen in dogs with congestive heart failure (CHF).” This should be rewritten as: “… as previously documented by other authors in dogs with congestive heart failure (CHF).” Then, you should provide pertinent references. Given the above, I think that the introduction of this study should be significantly modified by Authors.

- Response: References were added accordingly.

____

8 – Comment: This is wrong: all the numbers you are reporting are, as a matter of facts, an anticipation of results. In the M&M section you should simply state your methodology. Then, in the Result section, you will provide information of the number of dogs from each stage, their age, body weight, etc…

- Response: Thank you  for your appointment. We agree with the suggestion and deleted the sentence from M&M section, rewriting it by providing more information about the animals and placed it in results section as written: “The study population consisted of 60 animals both sexes (63.3% females and 36.7% males in group B1 and 43.3% females and 56.7% males in group B2). They consisted in 26 mixed breed dogs, 7 poodles, 6 dachshunds, 4 lhasa-apsos, 4 yorkshire terriers, 3 miniature pinschers, 2 malteses, 2 shih-tzus, 2 schnauzers, 1 australian cattle dog, 1 basset hound, 1 jack russel terrier and 1 labrador retriever. Ages ranged from 1.8 to 17 years were evaluated, and 30 dogs in stage B1 and 30 dogs in stage B2 were selected.

___

9 – Comment: Please, correct rewriting in this way: “Dogs were classified according to the criteria described in guidelines published by the American College of Veterinary Internal Medicine (ACVIM) guidelines in 2019 [15] (echocardiographic changes compatible with MMVD, systolic murmur in mitral focus ≥ III/VI, left atrium/aorta ratio ≥1.6 and left ventricle diastolic diameter normalized ≥1.7). Then, at the end of this sentence, add the proper reference (Keene et al., 2019).

- Response: We have accepted the suggestion and rewritten the sentece as: “Dogs were classified according to the criteria described in guidelines published by the American College od Veterinary Internal Medicine (ACVIM) guidelines in 2019 [15] (echocardiographic changes compatible with MMVD, systolic murmur in mitral focus ≥ III/VI, left atrium/aorta ratio ≥1.6 and left ventricle diastolic diameter normalized ≥1.7) [15].

____

10 – Comment: Please rewrite as: “Regarding drug administration, none of the enrolled dogs used cardiovascular therapy for the treatment of MMVD at the time of electrocardiographic analysis. In the case of stage B1 dogs, this was due to the fact that no medical therapy is recommended at this stage (Keene et al., 2019). In the case of stage B2 dogs, this was due to the fact that electrocardiographic data were acquired at the first presentation of the dogs, before the prescription of pimobendan (Keene et al., 2019). *(I hope to have correctly understood your methodology and what really happened).

- Response: You have perfectly understood, thank you for your consideration. The sentence was reviewed as follows: “Regarding drug administration, none of the enrolled dogs used cardiovascular therapy for the treatmend of MMVD at the time of the electrocardiographic analysis. In the case of stage B1 dogs, this was due to the fact that no medical therapy is recommended at this stage [15]. In the case of stage B2 dogs, this was due to the fact that electrocardhigraphic data were acquired at the first presentation of the dogs, before the prescription of pimobendan [15].

___

11 – Comment: Please rewrite as: “Exclusion criteria dogs with MMVD in stages different from B1 and B2, dogs with cardiac diseases different from MMVD, dogs with clinically relevant systemic diseases (e.g., infectious diseases presence of moderate-to-severe azotemia [i.e., serum creatinine >2.8, according to IRIS classification]), and dogs receiving vasoactive drugs and/or sedatives at the time of data collection.” Additionally, the abbreviation “IRIS” is used directly, which represents a mistake. Moreover, nart to the cut-off proposed by IRIS you should use the proper reference.

Response – The sentence was rewritten accordingly the orientations, as follows: Exclusion criteria dogs with MMVD in stagens different from B1 and B2, dogs with cardiac diseases different from MMVD, dogs with clinically relevant systemic diseases (e.g., infectious diseases, presence of moderate-to-severe azotemia [i.e., serum creatinine >2.8, according to International Renal Interest Society classification][16]), and dogs receiving vasoactive drugs and/or sedatives at the time of data collection”. IRIS website was also added to the references.

___

12 – Comment: Line 116: please, introduce at least one reference. I think that references should be added also at the end of sentences at lines 122, 126 and 133. In this way, readers will know which are the references/studies on which you have based your methodologies/measurements.

- Response: Thank you. I have added a book (Santilli, R. Electrocardiography of the dog & cat, 2nd ed, Editorial Edra, 2018) and two papers (DOI: 10.1515/pjvs-2015-0040 and 10.3390/ani10101829) for reference, as requested.

___

13 – Comment: Line 117: please, specify the degree of expertise of the examiner (was an ACVIM or ECVIM cardiology Diplomate? Not a board-certified cardiologist, but he/she holds a PhD or a MS in cardiology? Neither a board-certified cardiologist nor a vet with a PhD/MS in cardiology, but he/she is anyway a vet with many years of specilistic experience in small animal cardiology? Or was he/she a general vet?).

- Response: The examiners that performed the echocardiography and interpreted the Electrocardhigraphy in this study were members of the postraduation program in Veterinary Medicine from “Universidade Estadual Paulista “Júlio de Mesquita Filho” – UNESP Campus Botucatu”, from the Cardiology Department. We have added the information in the text, as requested: “The measurement of electrocardiographic intervals was made in a short duration (5 minutes) stretch, and  were manually repeated by a single examiner, member of the postraduation program in Veterinary Medicine from “Universidade Estadual Paulista “Júlio de Mesquita Filho” – UNESP Campus Botucatu”, from the Cardiology Department.

____

14 – Comment: Lines 142-143:  I think that this statment should appear at the beginning of M&M, not at its end (as usually done in other similar studies from canine cardiology).

- Response: Thank you for your reminder. The information was placed at the beggining of the M&M as requested.

___

15 – Comment:

- Additional comments:

1) Lines 123-126: I’m not sure to have completely/correctly understood your methodology. I read that you have mesured P and QT dispersion in each one of the six leads. Then my question is: which measurements did you considered in you analysis? Exclusively the ones from DII (as this lead is the one more used in canine cardiology)? Or did you perform a mean of values obtained in each lead? Or, alternatively, did you used simply the maximal value (regardless the lead from which it was obtained) instead of performing a mean?

- Response: We answered those questions by providing more information in the paragraphr, rewritting it as follows: “For each animal, the durations of P wave and QT intervals were obtained in 6 leads (D1, D2, D3, aVR, aVL, aVF), and then, maximum and minimum durations among these values were obtained. The dispersion was calculated using the formula d = max-min (Pd = Pmax-Pmin and QTd = QTmax – QTmin).” Please let us know if there are any more questions in this topic.

2) Even if Authors have focused their analysis on the study of very specific ECG parameters, I think that it would be very important to provide also “basic”/”conventional”  ECG data from the present study population (both from stage B1 dogs and stage B2 ones). Additional data should include all conventional ECG parameters, namely:

- duration, amplitude and mean electrical axis of P wave;

- duration of PQ interval;

 - duration and mean electrical axis of QRS complex, and amplitude of R wave;

- presence/absence of ST segment deviation and its amplitude;

- duration of QT and QTc interval;

- duration, polarity and amplitude of T waves.

Report mean/median of each parameter for B1 and B2 dogs along with pertinent reference intervals Of note, there are many books that provide reference ranges for many parameters. For those typically not reported by books, you can use papers available in t he current canine literature (e.g., P wave mean electrical axis = doi. 10.1016/j.jvc.2022.05.001 // ST segment = doi: 10.1111/jsap.13532. // T wave = doi:10.1016/j.jvc.2022.06.003.). Than, perform a statistical comparison of results from B1 and B2 dogs. All this will be certainly useful for readers to start to see/understand if some ECG difference exists between B1 and B2 dogs, before focus on more advanced ECG parameters.

3) There is also another lacking information. Did you noted also if, over the ECG recording, some of the enrolled dogs showed some disturbance of cardiac rhythm (e.g.,

a VPC, an APC or a second-degree AVB)? Or were all dogs compltely free from any arrhythmias duuring data recording? Please specify this in the manuscript: in the M&M

you should specify if you noted the possibile presence of these abnormalities; in the Rsult section, you should report the numeber of dogs showing thm and which arrhythmias you identified.

- Response: We appreciate your appointments. Unfortunately, not all data requested above was available for this study, however, information regards arrhythmic events ane standard ECG measurements was added (as requested by multiple reviewers) in the results section, as follows: When we consider standard electrocardiography measurements from one single beat [17], and heart rate (HR) measured on TEB ECG-PC VET® device, only QRS amplitude in mV showed statistically significance between groups (Tables 5 and 6), showing higher values in group B2.

The base rhythm was sinus arrhythmia in most pacients (25 dogs in group B1 and 24 dogs in group B2), with sinus rhythm occuring in 5 dogs from group B1 and 6 from group B2. 6 dogs from group B1 and 7 dogs from group B2 had sinus arrests;  2 dogs from grou B1 had ventricular premature complexes; 1 dog from group B1 and 5 from group B2 had supraventricular premature complexes; 1 dog from group B1 had righ bundle branch block and 2 dogs from group B2 had interatrial and/or intraatrial block [17] (Table 7)”.

____

16 – Comments: This sentence should be rephrase as I think that it will be very difficult to be understood by readers.

- Response: Thank you for the reminding: The sentence was reformulated as follows, in order to be fully understandable: “Regarding HRV parameters, the means and medians of RR intervals were higher in group B1 than in group B2. As the intervals between beats gets larger, the heart rate (HR) values decreases, and so, group B1 had a smaller HR than in group B2.”

____

17 – Comment: Lines 171-177: I my opinion, it is not essential to discuss data that did not reach the statistical significance. Therefore, you can delete all statement from this paragraph apart thie last one. You can rewrite it in this way: “Despite some mild discrepancies in the values of means and medians, it was not possible to observe any statistically significant difference related to HVR parameters (p <0.05)”.

- Response: We agree on this statement and rewrited the sentence as: “Despite some mild discrepancies in the values of means and medians, it was not possible to observe any statistically significant difference related to HRV parameters (p <0.05) (Table 4).”

____

18 – Comment: There is also another differential diagnosis. Please addi it. Moreover, at least a reference should be added at the end of this sentence. I suggest a book and also this manuscript: doi: 10.1111/j.1748-5827.2012.01200.x.

- Response: Thank you for this appointment. We have researched the paper suggested here and decided to delete the previous sentence from the discussion. We added the information, as follows: “P wave duration isolately, however, is known to be a limited parameter to evaluate atrial enlargement”.

____

19 – Comment: There is a typing error at the end of the sentence regardin reference n. 17.

- Response: Thank you for your appointment. Corrected as “These findings corroborate what was observed in the literature when compared to a recent study (2018), which evaluated the instability of the QT interval in patients with MMVD and observed a progressive increase in the indices as the disease progressed [17].”

___

20 – Comment - You should provide a reference, otherwise readers cannot understant which is the study published in 2021 you are referring to.

Response: Thank you for your appointment. We have decided to remove this paragraph.

____

21 – Comment: This is a very long sentence. I suggest you to rephrase it and break it into 3 different sententences to improve its clarity.

Response: We agree on this statement and rewritted as follows: The values of the HRV analysis of both groups were similar to those observed by Martinello et al. (2022) in healthy animals of similar ages. That allows us to infer that, alone, the presence of cardiac remodeling in MMVD was not able to change the sympathetic-parasympathetic balance in dogs without CHF. The same is observed in studies that analyzed dogs with CHF secondary to MMVD, such as the study by Oliveira et al. (2012), which observed a reduction in SDANN and pNN50 parameters in the group of animals in stage C.

___

22 – Comment: There is a typing error at the end of the sentence regardin reference n. 20.

Response: Thank you for your appointment. Corrected as: “In the same study, however, the analysis in the frequency domains showed a significant difference between B1 and B2 animals, obtaining a reduction in the HF vagal component and an increase in the LF/HF ratio in B2 in relation to B1[20].”

___

23 – Comment: As yor analysis did not reach the statistical significance, you cannot state this. Please, delete this sentence.

- Response: We agree on this statement and removed  the whole sentence from the study.

___

24 – Comment:  Additional comment for the entire Discussion: as said before, in each sentence and for each reference, it is important to specify which species you are referring to (so that readers can know wheter the findings you are describing is from canine or human literature).

Response: We agree on this statement and added information about human medicine in references 6, 10, 12, 13 and 14.

___

25 – Comment : I think that, in addition to the limitations already reported by the Authors, other relevant limitations should be listed, including:

1) the very limited number of dogs in each group;

2) the lack of a control group composed of 30 healthy dogs (to document whether B1 and/or B2 dogs have differences compared to healthy ones regarding the parameters examined by the authors);

3) the lack of intra- and inter-observer variability analysis concerning the measurements of examined advanced ECG parameters; and  

4) above all, the lack of 24-hour Holter monitoring (to obtain additional and more appropriate data on HRV parameters). Indeed, it is well known that, for many technical

and physiological reasons, HRV based on Holter recordings (especially if the Holter monitoring lasts for ≥ 20 hours, is free from artifacts, and includes recordings from the

night) provide a more precise/realistic picture of the autonomic balance of the patient compared to the one that cane obtained by a simple ECG performed over a few minutes

in the hospital setting.

And

26 – Comment: An additonal comment related to limitation is associated with the following sentence: “Regarding stage B2 patients, it is possible to observe a wide range of phenotypes, ranging from dogs with signs of cardiac remodeling to dogs about to present congestive heart failure (the latter, patients evolving to stage C). The different sizes of cardiac chambers as well as the different hemodynamic status turn B2 patients

into a particular group where electrophysiological changes may manifest in different ways as the disease progresses. Therefore, to precisely determine the beginning of the

alterations, further studies are needed to subclassify the B2 group and correlate electrophysiological changes to the severity of the disease”.

You included this section in the Conclusions. I believe it would be more appropriate to

delete it from the Conclusions and instead discuss it in the Limitations section. Additionally, you should include among your limitations the fact that you did not provide data on the echocardiographic parameters of your dogs. These data would have been useful to address the aforementioned point. Indeed, echocardiographic data would have provided information on the severity and heterogeneity of your population of B2 dogs. This could have been achieved simply by providing data on the enlargement of left-sided cardiac chambers among B2 dogs (e.g., LA/Ao ratio, LAD, LA volume), the severity of their left ventricular filling pressures (e.g., peak E wave, E/A ratio, E/IVRT, E/E' ratio), and the possible presence and degree of post-capillary pulmonary hypertension (e.g., by measuring the peak velocity of tricuspid regurgitation).

Response – Thank tou so much for your precious  appointments. The topics mentioned above were considered and discussed as limitations, as follows:

 “Another limitations regarding the retrospective  nature of this study is that only a few number of patients had fully cardiologic stantardized evaluation enough to fulfill the inclusion criteria, and for that reason, only a small number of dogs were included. This is also the reason why the creation of a control group and analysis of larger ECG tracings (i.e., 24 hour Holter monitorization) were no longer possible, since patients that are healthy usually does not perform a complete cardiologic standardized evaluation.

Another limitation concerning this study is that the electrocardiographic tracings were reviewed by a single examinator, causing a lack of inter-observer analysis.  The lack of statistics among echocardiographic exams is also a limitation observed.

Lastly, regarding stage B2 patients, it is possible to observe a wide range of phenotypes, ranging from dogs with signs of cardiac remodeling to dogs about to present congestive heart failure (the latter, patients evolving to stage C). The different sizes of cardiac chambers as well as the different hemodynamic status turn B2 patients into a particular group where electrophysiological changes may manifest in different ways as the disease progresses. Therefore, to precisely determine the beginning of the alterations, further studies are needed to subclassify the B2 group and correlate electrophysiological changes to the severity of the disease.”

___

27 and 28 – Comment -  Please delete from this section the part discussed above. Moreover, your conclusions should be overall changed in my opinion. Primarily, your conclusions should be significantly reduced, as they are very long (also because in this section you have repeated many concepts already discussed in previous sections of the manuscript). Usually, the Conclusions section is expected to be concise (e.g., 4-5 lines) and provide a very brief summary of the main findings of the study, nothing more.

Response: Thank you for the appointment. The sentence was relocated as requested, and the conclusion was rewrited as: “The reduction in HRV indices described in the literature in patients with stage C MMVD is not observed to the same extent in patients with stage B MMVD. The same reasoning is possible in relation to the parameters of P wave and QT interval dispersion; however, in contrast, when analyzing the instability of the QT interval, it is possible to observe an increase in STI, demonstrating the beginning of changes in the homogeneity of ventricular electrical activity reported in later stages.

____

29 – Comment -I think the list should be extended, at least by adding the study cited before.

Response – We agree on this statement and the study mentioned was included. Thank you for your recommendation.

Reviewer 2 Report

Comments and Suggestions for Authors

The authors investigated the heart rate ECG profile differences between MMVD dogs in the B1 and B2 stage.  This is an interesting study, but I do have a few questions:

1.      in line 13, authors state that MMVD is a slowly progression disease, I would just qualify the statement by saying that the progression is breed specific, and certain breeds, i.e. cavaliers, can progress much faster than others.

2.      Throughout the manuscript, the authors state that there is no cardiac remodeling in B1 but only in B2.  I do not think this is correct.  The B2 category includes heart sizes with significant cardiac enlargement to warrant pimobendan administration, not a delineation between having cardiac remodeling vs. no remodeling.  In fact, there are many dogs in the B1 category with larger than normal LA but just not big enough to meet the criteria for starting pimobendan.

3.      Line 15 talks about the cardiac silhouette.  This is a radiographic specific term and may not be most appropriate here.

4.      Line 29 states that the sympathovagal balance is the determinant for predisposition of arrhythmia.  This is not exactly accurate.  The sympathovagal balance determines the heart rate, but you need other factors for arrhythmia generation, i.e. cardiac fibrosis, electrolyte imbalance, etc.

5.      I understand that this is a retrospective study, but a 5 min recording of ECG is a major limitation to detect important changes in the heart rhythm.  For example, I would be most interested in seeing the analysis of actual arrhythmias.  Do these dogs have ventricular or supraventricular arrhythmias?  This would be a true gauge of the arrhythmogenic characteristics of these dogs.

6.      Line 127 states that the QT interval has been corrected.  What formula was used?

7.      Would analyzing the STI or LTI with correct QT change the results?

8.      It would be helpful for the authors to discuss the significance of STI more as it is not a commonly known concept to all, which will help the readers understand what a difference in STI would mean.

Comments on the Quality of English Language

 In general, the English writing will need additional editing

Author Response

Responses to reviewer 2:

1 - Comment: In line 13, authors state that MMVD is a slowly progression disease, I would just qualify the statement by saying that the progression is breed specific, and certain breeds, i.e. cavaliers, can progress much faster than others.

Response: Thank you for your gentile reminder. The sentence was rewritten as: “Myxomatous mitral valve disease (MMVD) is the most common cardiac disease in dogs. It affects the mitral valve apparatus and progresses slowly with age. In some breeds, especcially small dogs, i.e., cavaliers, the disease seems to progress faster than in other breeds”

____

2 – Comment: Throughout the manuscript, the authors state that there is no cardiac remodeling in B1 but only in B2. I do not think this is correct. The B2 category includes heart sizes with significant cardiac enlargement to warrant pimobendan administration, not a delineation between having cardiac remodeling vs. no remodeling. In fact, there are many dogs in the B1 category with larger than normal LA but just not big enough to meet the criteria for starting pimobendan.

Response: We agree on this statement. The sentence was rewritten, also considering the other reviewer’s statements, as follows: “According to the guidelines published by the American College of Veterinary Internal Medicine, canine MMVD is categorized into four stages. Dogs that are affected by MMVD but have never developed clinical signs of heart failure belong to stage B. This stage is further divided into stage B1 and stage B2 based on the dimensions of the left-sided cardiac chambers. Specifically, stage B1 is assigned if the dimensions of the left-sided cardiac chambers are not remodeled enough to require medical treatment. When they become enlarged enough to need therapy, the disease progresses to stage B2.”

___

Comment 3 -  Line 15 talks about the cardiac silhouette. This is a radiographic specific term and may not be most appropriate here.

- Response: Thank you for your reminder. The term was removed, and the sentence rewritten as follows: Therefore, the present study aimed at investigating whether an increase in the dimensions of the left-sided cardiac chambers in dogs with preclinical MMVD (stage B2) would predispose dogs to cardiac arrhythmias and/or autonomic imbalances. The comparison of electrocardiographic findings from stage B1 and stage B2 dogs showed no statistically significant predisposition to arrhythmias in subjects with enlargement of left-sided cardiac chambers. Moreover, the autonomic balance remais adequate in stage B2 as well as in stage B1

_____

Comment 4 - Line 29 states that the sympathovagal balance is the determinant for predisposition of arrhythmia. This is not exactly accurate. The sympathovagal balance determines the heart rate, but you need other factors for arrhythmia generation, i.e. cardiac fibrosis, electrolyte imbalance, etc.

- Response: We have rewritten the sentence in order to clarify that  sympathovagal balance was unchanged (HRV analysis) and prediposition of arryhthmias were slightly affected (STI parameter of QT interval dispersion): “The findings showed that cardiac remodeling present in the B2 stage was not able to significantly alter the sympathovagal balance and showed little interference with the predisposition of arrhythmias in dogs with MMVD”.

____

Comment 5 - . I understand that this is a retrospective study, but a 5 min recording of ECG is a major limitation to detect important changes in the heart rhythm. For example, I would be most interested in seeing the analysis of actual arrhythmias. Do these dogs have ventricular or supraventricular arrhythmias? This would be a true gauge of the arrhythmogenic characteristics of these dogs.

- Response: We appreciate your appointments. Information regards arrhythmic events was added to the study accordingly.

_____

Comment 6 - Line 127 states that the QT interval has been corrected. What formula was used?

Response: Thank you for your reminder. The sistem used Bazzet’s formula: QTc = QT/ √RR, we added the information in the text as follows: “QT measurements were corrected (QTC) using TEB ECG-PC VET® software using Bazzet’s formula: QTC = QT/ √RR.”

_____               

Comment 7 - Would analyzing the STI or LTI with correct QT change the results?

Response letter – The statistical analysis was performed with both QT and corrected QT, as described in Table 3. Only STI from non corrected QT had a statistical significance.

___

Comment 8 -  It would be helpful for the authors to discuss the significance of STI more as it is not a commonly known concept to all, which will help the readers understand what a difference in STI would mean.

We add reference number 16 that explains the importance of its verification in dogs with degenerative valve disease in stages B1 and B2

Reviewer 3 Report

Comments and Suggestions for Authors

Summary

This study compares heart rate variability in 60 dogs with stage B1 and B2 myxomatous mitral degenerative valve disease. The present study investigated whether stage B2 dogs have a greater predisposition to cardiac arrhythmias. The authors conclude that there was “no predisposition to arrhythmias during this stage of the disease, and the autonomic balance remains adequate.”

Comment

1.       This is a potentially interesting study that examines the progression of mitral valve disease. Several weaknesses need to be addressed.

2.       The authors need a control group of normal dogs. It is not clear whether the HRV of B1 and B2 are different than control values.

3.       The author implies that the classification was also made based on echo data. These data should be included.

4.       Demographic data, like weight should be included. The term “small dog” is ambiguous. It needs to be quantified.

5.       Since the authors calculated the LA/Aorta ratio they should include this data in the tables along with the LV size. In fact, in the abstract, they state that “cardiac silhouette” may be a predictor variable. They need to quantify this data.

6.       In the HRV, they omit HR rate data. (mean, maximum and minimum)

7.       Again, it is necessary to compare all the data to a normal control group. 

Comments on the Quality of English Language

none

Author Response

Responses to reviewer 3:

1 - Comment :  This is a potentially interesting study that examines the progression of mitral valve disease. Several weaknesses need to be addressed.

Response: Thank you so much.
____

2 – Comment:  The authors need a control group of normal dogs. It is not clear whether the HRV of B1 and B2 are different than control values.

Response: Thank you so much for your valuable appointment. In our revision, a similar study from Bruller et. al., 2018 (DOI: 10.1016/j.jvc.2018.06.002) did not used a control group. We are definitely considering this suggestion for further studies, however, in this present data only stage B dogs were evaluated.

____

3 – Comment: . The author implies that the classification was also made based on echo data. These data should be included.

Response: Thank you for your observation. The information about left atrial/aorta ratio and left ventricle size from the patients were used to classify those patientes, however, were not statistically analyzed. That is the major reason why they were not included on this paper. All patients were, however, classified accordingly ACVIM consensus (Keene et. al, 2019). We are aware of your valuable consideration about this data, and this appointment will definitely be considered for further studies. 

___

4 – Comment: . Demographic data, like weight should be included. The term “small dog” is ambiguous. It needs to be quantified.

Response: Thank you for your observation. Although our study was mased usind stored data, some data about weight were not available. In this study we used data from dogs from different breeds, including larger dogs. We have added the information about the breeds in the results section as follows: “The study population consisted of 60 animals both sexes (63.3% females and 36.7% males in group B1 and 43.3% females and 56.7% males in group B2). They consisted in 26 mixed breed dogs, 7 poodles, 6 dachshunds, 4 lhasa-apsos, 4 yorkshire terriers, 3 miniature pinschers, 2 malteses, 2 shih-tzus, 2 schnauzers, 1 australian cattle dog, 1 basset hound, 1 jack russel terrier and 1 labrador retriever. Ages ranged from 1.8 to 17 years were evaluated, and 30 dogs in stage B1 and 30 dogs in stage B2 were selected.”
_____

5 – Comment:. Since the authors calculated the LA/Aorta ratio they should include this data in the tables along with the LV size. In fact, in the abstract, they state that “cardiac silhouette” may be a predictor variable. They need to quantify this data.

Response: Thank you for your observation. We have removed the term “cardiac silhouette”,  as others reviewers pointed out that this is a radiology term, and radiographic images were not  used in this study. About LA/AO ratio and left ventricle sized, as stated before, were used to classify the patients and were not statistically analyzed, and for this reason, were not included in this paper.

____

6 – Comment:  In the HRV, they omit HR rate data. (mean, maximum and minimum)

Response: The mean HR and maximum HR were not statistically analyzed, however, the data “MEAN HR”, mentioned in the results section is present in Table 4 as indicated in the reformulated paragraph, as follows: “Regarding HRV parameters, the means and medians of RR intervals were higher in group B1 than in group B2. As the intervals between beats gets larger, the heart rate (HR) values decreases, and so, group B1 had a smaller HR than in group B2. Despite some mild discrepancies in the values of means and medians, it was not possible to observe any statistically significant difference related to HRV parameters (p <0.05)(Table 4.)
___

7 – Comment:  Again, it is necessary to compare all the data to a normal control group.

- Response: Thank you so much for your appointment. Limitations about the absence of a control group were discussed as follows: “Another limitations regarding the retrospective  nature of this study is that only a few number of patients had fully cardiologic stantardized evaluation enough to fulfill the inclusion criteria, and for that reason, only a small number of dogs were included. This is also the reason why the creation of a control group and analysis of larger ECG tracings (i.e., 24 hour Holter monitorization) were no longer possible, since patients that are healthy usually does not perform a complete cardiologic standardized evaluation.” 

Round 2

Reviewer 1 Report

Comments and Suggestions for Authors

Comment to authors are present in the attached file.

Comments on the Quality of English Language

NA

Author Response

LETTER RESPONSE TO EDITOR AND REVIEWERS

Dear Editor and Reviewers;

We appreciate your considerations. The most recent modifications have been highlighted in blue in the manuscript.

Responses to reviewer:

1 - Comment: -line 13: change “cavaliers” to “Cavalier King Charles Spaniels”

-  Response: Revised accordingly. ____

2 - Comment:  - line 86: delete “requiring treatment”

- Response: The sentence was not corrected, since the previous reviewer correctly stated that  B1 dogs, indeed, may have cardiac remodeling (i.e. atrial enlargement without ventricle enlargement).

The sentence was revised as follows: “Because changes in myocardial structure and HRV reduction in dogs with MMVD predispose them to arrhythmias, the present study aims to compare the associated markers of P-wave and QT interval dispersion, QT interval instability, and HRV analyses in asymptomatic MMVD dogs with and without cardiac remodeling that recquires treatment (stages B2 and B1, respectively).”

____

3 - Comment:  - Lines 116-118: The following statement does not effectively clarify the evaluator's expertise in cardiology (“This evaluator is a veterinarian, a member of the postgraduate program in veterinary medicine at Universidade Estadual Paulista ‘Júlio de Mesquita Filho’—UNESP Campus Botucatu, in the Cardiology Department”). Instead, it merely indicates where the colleague works. You should specify whether the veterinarian who performed the ECG analysis (on which the manuscript is based) is a board-certified ACVIM or ECVIM cardiologist, or if he/she is not a cardiology specialist. If the latter, you should clarify whether the colleague, despite not being a specialist, has extensive experience in small animal cardiology (for example, in some studies, it is possible to find statements like: “analyses were performed by a professor of veterinary internal medicine with more than 15 years of experience in small animal cardiology”). Such statements help to clarify the qualifications of the person conducting the analysis.

- Response: This study was performed during the Master’s Degree Program. The examiner (this autor)  was, at the time, a veterinary with a residency in Small Animal Medicine that had 2 years of training in ECG during the residency program. As requested, the sentence was revised as follows: This evaluator is a veterinarian, a member of the postgraduation program in Veterinary Medicine in the Cardiology Department (a veterinary with a residency in Small Animall Medicine and two years of experience in electrocardiography).

________

4 Comment –  -line 132: “The electrocardiographic intervals” Which intervals? Spease, specify in this sentence.

Response: The paragraph wa reformulated as follows: “Computerized electrocardiographic exams were recorded in 3 minutes. The exam was performed in unsedated dogs positioned and restrained in right lateral recumbency, following the technique described by Tilley in 1992 [19]. The electrodes were attached to the humerus–radius–ulna and femur–tibia–patella joints, and damped in alcohol 70% solution. The bipolar (DI, DII, and DIII) and unipolar (aVR, aVL, and aVF) leads were recorded and processed by TEB ECG-PC VET® software. Standard electrocardiography measurements (P wave duration and amplitude, PR interval duration, QRS duration and amplitude, QT interval duration, T wave amplitude and RR intervals) were measured in a single beat and analyzed accordingly to Santilli et al [20]”.

_____

5 Comment - lines 133-135: if the operator is the same, here you should not repeat all the pertinent description, but simple state that these analyses were performed by the same oprator.

Response: The sentence was deleted from the new paragraph, as mentioned above.

___

6 Comment –  -In the Results section, it now present Table 5, a table where is possible to see the results of the analysis of standard ECG measurments. However, in the M&M section there are no information of the way these measurmenges were made. In many studies on ECG, there are detailed descriptions similar to the following one:

“In all dogs, an electrocardiogram was conducted with the dog positioned and manually restrained in right lateral recumbency, following the technique described by Tilley in 1992 [Tilley LP.Principles of electrocardiographic recording. In: Tilley LP, editor. Essentials of Canine and Feline Electrocardiography: Interpretation and Treatment. 3rd ed. Philadelphia: Lippincott Williams & Wilkins; 1992, p. 21–39.]. All electrocardiograms were recorded in unsedated dogs using a commercially available machinese. For each animal, an effort was made to obtain an electrocardiographic tracing showing a clean baseline with easily recognizable waveforms. The same investigator manually measured intervals and amplitudes using a caliper and ruler with 0.5-mm graduations. Three representative consecutive beats were used to measure various electrocardiographic variables, and the results were averaged for each variable. Initially, the heart rhythm and conventional variables (i.e., heart rate; amplitude, duration and mean electrical axis in the frontal plane [MEA] of the P wave; PQ interval duration; R wave amplitude; duration and MEA of the QRS complex; ST segment amplitude; duration, amplitude and MEA of T wave, and duration fo QT interval and QT interval corrected fot the heart rate) were assessed according to the standard technique to evaluate if there were electrocardiographic abnormalities [Santilli RA, Moïse NS, Pariaut R, Perego M. Formation and interpretation of the electrocardiographic waves. In: Santilli RA, Moïse NS, Pariaut R, Perego M, editors. Electrocardiography of the Dog and Cat – Diagnosis of Arrhythmia. 2nd ed. Milano: Edra; 2018, p. 35–70. //// Oliveira P. Cardiac vectors and the genesis of the electrocardiogram. In: Willis R, Oliveira P, Mavropoulou A, editors. Guide to Canine and Feline Electrocardiography. 1st ed. Oxford: John Wiley & Sons; 2018, p. 21–33.]. Such variables were judged to be normal/abnormal according to generic canine RIs [ref on general reference ranges = Santilli RA, Moïse NS, Pariaut R, Perego M. Formation and interpretation of the electrocardiographic waves. In: Santilli RA, Moïse NS, Pariaut R, Perego M, editors. Electrocardiography of the Dog and Cat – Diagnosis of Arrhythmia. 2nd ed. Milano: Edra; 2018, p. 35–70. //// ref on general reference ranges = Oliveira P. Cardiac vectors and the genesis of the electrocardiogram. In: Willis R, Oliveira P, Mavropoulou A, editors. Guide to Canine and Feline Electrocardiography. 1st ed. Oxford: John Wiley & Sons; 2018, p. 21–33).

This is just an example on the way rigorous studies on ECG describe accurately the methodology employed. I suggest authors to do something similar. Basically, it is asked to be more provide a more precise and detailed description of the methodology, and to use additional referfences (from the ones above cited) for the variables analysed by authors.

Response: We agree on this statement and described the technique on the paragraph as mentioned above:

“Computerized electrocardiographic exams were recorded in 3 minutes. The exam was performed in unsedated dogs positioned and restrained in right lateral recumbency, following the technique described by Tilley in 1992 [19]. The electrodes were attached to the humerus–radius–ulna and femur–tibia–patella joints, and damped in alcohol 70% solution. The bipolar (DI, DII, and DIII) and unipolar (aVR, aVL, and aVF) leads were recorded and processed by TEB ECG-PC VET® software. Standard electrocardiography measurements (P wave duration and amplitude, PR interval duration, QRS duration and amplitude, QT interval duration, T wave amplitude and RR intervals) were measured in a single beat and analyzed accordingly to Santilli et al [20].”

______

7 Comment -In the current version of the Result section there is a description of arrhythmias documented along with a new Table (“The base rhythm was sinus arrhythmia in most patients (25 dogs in group B1 and 24 dogs in group B2), with sinus rhythm occurring in 5 dogs from group B1 and 6 from group B2. Six dogs from group B1 and seven dogs from group B2 had sinus arrests; two dogs from group B1 had ventricular premature complexes; one dog from group B1 and five from group B2 had supraventricular premature complexes; and one dog from group B1 had a right bundle branch block and two dogs from group B2 had interatrial and/or intra-atrial blocks [18]”). There are some problems with this. First, what is present in the text should noty be repeated in Tables. Therefore, I suggest authors to select if present these data in the text or in Table 7. Second, you should provide before in the M&M scetion a brief explanation on the fact, that in your study, you also looked for the possible presence of arrhythmias.

Response: Thank you for your appointment. The table was deleted from the text. As requested, we now added the sentence:  “Because this study aims to evaluate the arrhythmogenic profile of B stage dogs, all of the arrhythmic events observed during the monitorization in both groups were described.”

_____

8 Comment - *Additional comment: In the current version of the manuscript, information and tables on advanced ECG analysis appear first, while data and tables on standard ECG analysis are presented later. This seems illogical to me. I suggest the authors change the order in both the Materials & Methods and Results sections, so that readers first encounter the standard ECG measurements, followed by the presence of arrhythmias, and finally the advanced ECG analysis.

Response: As requested, simple ECG measurements were placed before in the results section, followed by more specific measurements.

____

9 Comment - According to Table 5, it seems that some standard ECG paramters have not been measured, namly the mEA of P wave, the ST segemnt amplitude, the duration of the QT interval corrected for heart rate, the T wave duration and MEA. This should be added to the list of limitations, as these parameters have been extensively studied in dogs and pertinent reference intervals are present (ref for P wave MEA = doi. 10.1016/j.jvc.2022.05.001 // ref for ST segmebt = doi: 10.1111/jsap.13532. // ref for T wave = doi: 10.1016/j.jvc.2022.06.003.). I suggest the authors include this limitation along with the aforementioned references to make this point clear to readers and to ensure that future studies on ECG measurements are not lacking a complete data set of ECG parameters.

Response: We agree on this statement and limitations were described as follows: “Due to the fact that this study focused in more specific analysis, some standard ECG parameters were not statistically evaluated, such as the mean electrical axys of P wave, the ST segment amplitude, the T wave duration and mean electrical axys of the QRS complex.”

The QTc, however, was used in this study as it is part of the QT instability analysis.

____

10 -  Comment -  Table 5: please, add a column to provide reference intervals for each ECG parameter along with the pertinent reference, so that readers will be able to undestrand if there are some measurements above/below normality ranges.

Response: The parameters were added accordingly to reference 20 in table 2.

_____

11- Comment - Please, expand your reference list as suggested during the revision.

Response: The reference for electrocardiographic tecnique was added accordingly.

Reviewer 3 Report

Comments and Suggestions for Authors

None

Author Response

WE UNDERSTAND THAT THERE ARE NO CONSIDERATIONS TO BE ANSWERED ACCORDING TO THIS REVIEWER. THANK YOU FOR YOUR ATTENTION.